# Reduction in Maternal Energy Intake during Lactation Decreased Maternal Body Weight and Concentrations of Leptin, Insulin and Adiponectin in Human Milk without Affecting Milk Production, Milk Macronutrient Composition or Infant Growth

**DOI:** 10.3390/nu13061892

**Published:** 2021-05-31

**Authors:** Gabriela E. Leghi, Merryn J. Netting, Ching T. Lai, Ardra Narayanan, Michael Dymock, Alethea Rea, Mary E. Wlodek, Donna T. Geddes, Beverly S. Muhlhausler

**Affiliations:** 1School of Agriculture, Food and Wine, The University of Adelaide, Adelaide 5064, Australia; gabriela.estevesleghi@adelaide.edu.au; 2Women and Kids Theme, South Australian Health and Medical Research Institute (SAHMRI), Adelaide 5000, Australia; merryn.netting@sahmri.com; 3Discipline of Paediatrics, The University of Adelaide, Adelaide 5005, Australia; 4School of Molecular Sciences, The University of Western Australia, Perth 6009, Australia; ching-tat.lai@uwa.edu.au (C.T.L.); ardranarayanan1994@gmail.com (A.N.); donna.geddes@uwa.edu.au (D.T.G.); 5Centre for Applied Statistics, The University of Western Australia, Perth 6009, Australia; michael.dymock@uwa.edu.au; 6Mathematics and Statistics, Murdoch University, Perth 6150, Australia; alethea.rea@murdoch.edu.au; 7Department of Physiology, School of Biomedical Sciences, The University of Melbourne, Melbourne 3010, Australia; m.wlodek@unimelb.edu.au; 8Commonwealth Scientific and Industrial Research Organisation (CSIRO), Kintore Avenue, Adelaide 5000, Australia

**Keywords:** human milk, lactation, diet, macronutrients, hormones, leptin, adiponectin, insulin

## Abstract

Maternal diet has the potential to affect human milk (HM) composition, but very few studies have directly assessed the effect of maternal diets on HM composition. The primary aim of this study was to assess the effect of improving dietary quality in lactating women over 2 weeks on the concentrations of macronutrients and metabolic hormones in HM. The secondary aims were to assess the impact of the dietary intervention on 24 h milk production, maternal body composition and infant growth. Fifteen women completed a 1-week baseline period followed by a 2-week dietary intervention phase targeted towards reducing fat and sugar intake. Maternal anthropometric and body composition and infant growth measurements were performed weekly. Total 24 h milk production was measured before and after the dietary intervention, and HM samples were collected daily. Maternal intakes of energy (−33%), carbohydrate (−22%), sugar (−29%), fat (−54%) and saturated fat (−63%) were significantly reduced during the dietary intervention. HM insulin, leptin and adiponectin concentrations were 10–25% lower at the end of the dietary intervention, but HM concentrations of macronutrients were unaffected. Maternal body weight (−1.8%) and fat mass (−6.3%) were significantly reduced at the end of the dietary intervention, but there were no effects on 24 h milk production or infant growth. These results suggest that reducing maternal energy, carbohydrate, fat and sugar intake over a 2-week period is associated with significant reductions in HM insulin, leptin and adiponectin concentrations. These changes may be secondary to decreases in maternal weight and fat mass. The limited studies to date that have investigated the association between metabolic hormone concentrations in HM and infant growth raise the possibility that the changes in HM composition observed in the current study could impact infant growth and adiposity, but further studies are required to confirm this hypothesis.

## 1. Introduction

HM composition is uniquely species-specific and, as such, provides the optimal nutrition to support the growth and development of human infants [1]. However, the levels of essential nutrients, immune factors and hormones differ substantially between lactating women, and these differences may impact the short- and longer-term health of the child [2,3]. Emerging evidence suggests that the variation in HM composition may be related to maternal, geographical and other environmental factors [4,5,6].

One factor that has the potential to impact HM composition is variations in maternal diet; however, very few studies have directly assessed the effect of maternal diets on HM composition. The relationship between specific dietary components and the levels of HM components is also poorly defined; a systematic evaluation of the existing literature suggested that the only nutrients for which there was reasonable evidence of an association between maternal dietary intake and HM concentrations were individual fatty acids and vitamin C [7]. In addition, there is limited understanding of how changes to the maternal diet during lactation can impact HM macronutrient composition. The only study to date to directly investigate the impact of a dietary intervention on HM macronutrient composition reported that HM fat and total energy content were increased in lactating women who consumed a high-fat, low-carbohydrate (30% carbohydrate, 55% fat) diet for 8 days. In this same study, however, HM fat and energy content were not altered when women consumed a low-fat, high-carbohydrate (60% carbohydrate, 25% fat) diet for the same period, and HM production, lactose and protein concentrations were not affected by either of the dietary interventions [8]. It is important to note, however, that this study did not provide details of participants’ habitual diet and included women who were not exclusively breastfeeding, which is also known to influence HM composition [9]. To the best of our knowledge, no studies have investigated the effect of a maternal dietary intervention during lactation on HM concentrations of metabolic hormones.

Many women in Western countries consume poor-quality diets that contain excessive levels of added sugars and fat [10,11], and studies in experimental animal models suggest that this dietary pattern has the potential to impact the macronutrient, fatty acid and metabolic hormone composition of the mother’s milk [12,13,14]. However, no human studies have identified the effect of improving dietary quality during lactation on the composition of HM. Therefore, the primary aim of this study was to determine the effect of reducing energy, fat and sugar intakes in lactating women over a 2-week period on the concentrations of macronutrients and metabolic hormones in HM. The secondary aims of the study were to assess the effect of this dietary intervention on 24 h milk production, maternal body composition and infant growth. We hypothesised that the dietary intervention would result in significant changes in HM composition in the absence of any changes in milk production or infant intake.

## 2. Materials and Methods

### 2.1. Study Design

This study was an open label dietary intervention trial. The study design has previously been published in detail [15]. Briefly, after consuming their habitual diet for the 1st study week, women completed a 2-week dietary intervention phase that aimed to limit intake of total sugar by targeting added sugars, which include added forms of fructose, lactose, dextrose, sucrose, fruit and sugar syrups according to Food Standards Australia New Zealand [16]. The timeline for this proof-of-concept study was chosen based on findings from the limited previous studies that had evaluated the impact of dietary changes on HM composition. This study was approved by the Human Research Ethics Committee of the University of Western Australia (RA/4/20/4953) and registered with the Australian New Zealand Clinical Trials Registry (ACTRN12619000606189) on 23 April 2019.

### 2.2. Study Participants

Women were recruited via social media, existing networks and community centres. As published previously, to be eligible to participate, women had to be exclusively breastfeeding a term singleton infant between 6 and 20 weeks postpartum and growing normally (according to WHO standards) [15,17]. Exclusion criteria included pregnancy complications, maternal diabetes, maternal diseases known to affect gastric absorption, restrictive diets, maternal smoking or any major infant congenital abnormalities or health issues that could affect feeding behaviour.

### 2.3. Dietary Assessment

After enrolment, women completed three 24 h dietary recalls (two on weekdays and one on a weekend day) for 1 week before the dietary intervention using the Automated Self-Administered 24-Hour Dietary Recall (ASA24), which has been previously validated in pregnant women [18] and women of childbearing age [19] against measures of true intakes and interviewer-administered 24 h recall. Images assisted with portion size estimation, and food codes from the Australian Food Supplement and Nutrient Database (AUSNUT) 2011–2013 are automatically assigned.

### 2.4. Dietary Intervention

After the first week of the study, during which women were instructed to follow their habitual diet, women participated in a 2-week dietary intervention phase, which was aimed at improving dietary quality by reducing intakes of discretionary foods, saturated fats and added sugars. Women received all foods/snacks from a home delivery service and were asked to record any additional foods/drinks consumed as well as any of the provided foods that were not eaten. No women reported food allergies or intolerances that required modifications to the meal plan during the dietary intervention phase. A dietitian tailored a nutritionally balanced and portion-controlled 1553–1975 kcal diet (based on the participant’s energy requirements and personal preferences) according to the Australian Dietary Guidelines. Overall, the diets had at least a 500 kcal deficit compared to their estimated energy requirements, <22% of total energy from total sugar, <10% of total energy from saturated fat and > 30 g of fibre. The dietitian contacted women via their preferred means twice weekly to assess compliance with the diet. Total dietary intake per participant was calculated based on the daily consumption of provided diet and daily consumption of any extra food or beverages using the nutrition analysis software FoodWorks 10 Professional (Xyris Pty Ltd., 2019, Brisbane, Australia). Total sugar intake was defined as a sum of fructose, glucose, sucrose, maltose, lactose and galactose, while carbohydrate intake was defined as a sum of total sugars, maltotriose, starch and other available carbohydrates (glycogen + raffinose + stachyose + dextrins + maltodextrins + other undifferentiated oligosaccharides), as determined using FoodWorks 10 Professional (Xyris Pty Ltd., 2019, Brisbane, Australia).

### 2.5. Anthropometry Measurements

All maternal and infant measurements were performed immediately before and at 1 week and 2 weeks after the start of the dietary intervention (weeks 2 and 3). Maternal weight was measured using a calibrated electronic scale (accuracy ±0.1 kg; Seca, Chino, CA, USA). Height was measured against a calibrated marked wall (accuracy ±0.1 cm; Stadiometer). Infant weight was measured before breastfeeding using a calibrated electronic scale (accuracy ±2 g; Medela Inc., McHenry, IL, USA). Infant length and head circumference were also measured in accordance with previous published collection protocols [20]. Maternal and infant body mass index (BMI) were calculated as kg/m^2^. Infant growth measures (weight-for-length z-score, weight-for-age z-score and length-for-age z-score) were calculated according to WHO standards [21].

### 2.6. Body Composition Measurements

Maternal body composition was assessed immediately before (end of week 1) and at the end of 1 week and 2 weeks after the start of the dietary intervention (weeks 2 and 3), and all measurements (fat mass, percentage fat mass and fat-free mass) were performed in duplicate after a breastfeeding session (standard procedure in order to avoid interferences from variable milk volume). These collection methods using a ImpediMed SFB7 tetrapolar bioelectrical impedance analyser (ImpediMed, Brisbane, Australia), including calibration testing have been described previously [20]. Within-participant coefficient of variation (CV) for maternal percentage fat mass was 0.21%. 

### 2.7. Human Milk Sample Collection

The procedures for HM sample collection have previously been described in detail [15]. Briefly, women collected HM samples (~5 mL) daily before the first morning feed and at least 2 h after the previous feed throughout the study. Women also carried out intensive milk sampling once per week. The intensive sampling involved collecting HM samples from each breast at 3 separate timepoints across a 24 h period: before the first feed in the morning, one in the afternoon and one in the evening. As per the study protocol, the total number of HM samples collected across the study period for each woman was 54 samples (27 from each breast). A summary of HM sample collection per participant is provided in Table 1. For each sample collection, women were asked to collect prefeed samples aseptically from both breasts either by hand expression or by using a breast pump (either manual or electric). Participants stored the samples in their home freezer until weekly collection by study staff, after which they were transported to the analytical laboratory on ice and then stored at −80 °C until analysis.

### 2.8. 24 h Milk Production and Milk Intake

Mothers measured milk production and infant milk intake using a standardised 24 h milk production protocol [22,23] at 2 timepoints in the study—before (week 1) and after the dietary intervention (week 3). Briefly, they weighed their fully clothed infant on calibrated electronic scales (Medela Baby Scale, accurate to 2 g) before and after each feed from each breast for one 24 h period plus one additional breastfeed. Milk intake was calculated by subtracting the weight of the infant at the start of a feeding session from their weight at the end of the session and amounts of HM (g) consumed by the infant were recorded for each feeding session. The 24 h intake of milk was the sum of the volumes of milk received from the breast and expressed milk within a 24 h period (24 h = beginning of first feed to 24 h). Milk production included both milk removed by the infant and that expressed by the mother and may therefore be greater than the milk intake. It was calculated according to the equation $P=\sum_{t=1}^{N}Vt\frac{N-1}{N}\frac{24}{T}$, where one feed from each breast is recorded after the 24 h period [22,23]. Test weighing has been shown to be equivalent to the dose to mother deuterium oxide method for measuring milk production [24].

### 2.9. Calculated Intake of Human Milk Macronutrients

Milk intakes were defined as the amount of macronutrient ingested over a 24 h period. For protein and lactose, this was calculated by multiplying the average concentration of the respective macronutrient for the right and left breasts by the volume of milk consumed from each breast. The intakes from each breast were then summed to arrive at the 24 h milk intake (g). In the case for fat, intake was calculated by first averaging the pre- and postfeed fat concentrations for each feed and multiplying by the respective volume consumed by the infant for each feed. The sum of all feeds gave the fat intake for the 24 h period on the day that milk production was assessed. Separate milk samples were collected before and after every feed for the purpose of calculating total fat intake (more accurate than using only a prefeed sample) [25].

### 2.10. Sample Preparation

Whole milk (500 µL) samples were thawed and transferred into a homogeniser tube containing zirconium beads (3.0 mm diameter, D1032-30, Benchmark Scientific, Sayreville, NJ, USA). The milk samples were homogenised for 3 cycles at 2500 rpm, 5 s per cycle with a 3 s pause using a microtube homogeniser (BeadBug 6, Benchmark Scientific, Sayreville, NJ, USA). The assays for macronutrient and metabolic hormone measurements (all previously validated for use in human milk) were performed using a JANUS automatic pipetting workstation (PerkinElmer, Inc., Waltham, MA, USA) and measured on a EnSpire plate reader (PerkinElmer, Inc., Waltham, MA, USA).

### 2.11. Macronutrient Measurements

Measurements for concentrations of fat, protein and lactose in HM, including preparation of HM standards and recovery assays, have been previously documented in detail [15]. Briefly, fat concentration of whole milk samples (including milk samples from the 24 h milk production) was measured using a validated creamatocrit method [26]. The percentage fat content obtained using this method was converted to total fat content (g/L) using a validated equation (lipid = 3.968 + [5.917 × creamatocrit value]) [27]. 

Protein concentration in defatted milk samples was assessed using a modified Bradford method, as previously described [28]. Lactose concentration in defatted milk samples was determined in duplicate by an enzymatic method [29,30]. 

### 2.12. Metabolic Hormone Measurements

The concentration of adiponectin in the homogenised milk sample was determined in duplicate by the Biovendor human adiponectin ELISA kit (RD191023100, Lot E19-046, Biovendor-Laboratorni medicina a.s., Brno-Řečkovice a Mokrá Hora, Czech Republic). The homogenised whole milk was diluted 3-fold with a sample buffer in accordance with the manufacturer instructions. The recovery of a known amount of adiponectin added to the milk samples was 95 ± 7% (*n* = 7). The detection limit of this assay was 0.61 ng/mL, the inter-assay CV was 7% (*n* = 7) and the intra-assay CV was 4.49% (*n* = 220).

The concentration of insulin in the neat, homogenised whole milk sample was determined in duplicate by the Biovendor human insulin ELISA kit (RIS006R, Lot X19-342, Biovendor-Laboratorni medicina a.s., Brno-Řečkovice a Mokrá Hora, Czech Republic). The recovery of a known amount of insulin added to the milk samples was 97 ± 7% (*n* = 7). The detection limit of this assay was 4.88 µIU/mL, the inter-assay CV was 3% (*n* = 7) and the intra-assay CV was 8.05% (*n*= 160).

The concentration of leptin in the neat, homogenised whole milk sample was determined by the R&D systems leptin DuoSet ELISA kit (DY398, Lot P209801, R&D Systems Inc, USA). The recovery of a known amount of leptin added to the milk samples was 99 ± 1.7% (*n* = 7). The detection limit of this assay was 0.19 ng/mL, the inter-assay CV was 12% (*n* = 7) and the intra-assay CV was 5.48% (*n* = 220).

### 2.13. Statistical Analysis

The data were analysed by fitting a linear mixed-effects model to each of the response variables: fat, protein, lactose, leptin, insulin and adiponectin concentrations, 24 h milk production, anthropometry and body composition. Week, collection breast (left or right) and time of day were input as explanatory variables for each of the respective models. In all models, the mothers were fitted as random effects as to account for within-mother variation. Model selection was carried out using backwards selection. All statistical analyses, including descriptive statistics (mean, standard deviation (SD) and range)) were computed using the statistical programming language R [31]. A test was considered statistically significant when its *p*-value was <0.05.

## 3. Results

### 3.1. Study Participants

The participant flow for this study has previously been reported in detail [15]. Briefly, 66 women initially expressed interest in the study; 38 subsequently completed the eligibility assessment, of whom 18 were enrolled and 15 completed the study (Figure 1). Clinical and sociodemographic characteristics of the study participants (mothers and infants) have been reported previously [15] and are shown in Table 2.

A total of 796 HM samples were provided and analysed in duplicate for each of the macronutrients. Eleven women provided a complete set of 54 HM samples, and the remaining 4 women provided between 47 and 53 HM samples. Further details on the number of samples analysed for macronutrients can be found in a previous publication [15]. Of the 796 HM samples, 257 were selected for the assessment of metabolic hormones (18 samples per participant with 13 samples missing).

### 3.2. Dietary Intake before and during Dietary Intervention

The daily intakes of total energy and key macronutrients during the baseline week and dietary intervention period are presented in Table 3. Women consumed significantly less total energy (−33%; −828 ± 585 kcal), carbohydrate (−22%; −55 ± 38 g), sugar (−29%; −34 ± 24 g), fat (−54%; −62 ± 44 g) and saturated fat (−63%; −29 ± 20 g) during the dietary intervention phase compared to baseline. The intake of dietary fibre and percentage of energy intake from protein and carbohydrate were increased, and the percentage of energy from fat and saturated fat was decreased during the intervention phase.

### 3.3. Effect of the Dietary Intervention on HM Macronutrient and Metabolic Hormone Composition 

There were no changes in the concentrations of HM fat, protein and lactose in either the first or second week after commencement of the dietary intervention (Table 4).

The concentrations of insulin in HM were significantly lower during both weeks of the dietary intervention phase (−25%) compared to the baseline period. HM concentrations of leptin (−20%) and adiponectin (−10%) were also both significantly lower at the end of the dietary intervention (week 3 of the study) compared to levels at baseline (Table 4). 

### 3.4. Effect of the Dietary Intervention on HM Production and Intake of Macronutrients and Metabolic Hormones 

There was no significant effect of the dietary intervention (week 1 vs. week 3) on HM production (760 mL vs. 773 mL, *p* = 0.74) or the infants’ 24 h intakes of either fat (72 g/24 h vs. 78 g/24 h, *p* = 0.48), protein (6 g/24 h vs. 6 g/24 h, *p* = 0.42) or lactose (49 g/24 h vs. 48 g/24 h, *p* = 0.88). However, the dietary intervention did have a significant effect on infants’ 24 h intakes of both leptin (0.18 ng/24 h vs. 0.14 ng/24 = h, *p* = 0.017) and insulin (19.93µIU/24 h vs. 15.52 µIU/24 h, *p* = 0.020), but not on adiponectin (9.66 ng/24 h vs. 9.03 ng/24 h, *p* = 0.40).

### 3.5. Effect of the Dietary Intervention on Maternal Anthropometrics and Infant Growth

Maternal body weight (−1.8%, *p* = 0.002), BMI (−5.8%, *p* = 0.002), fat mass (−6.3%, *p* = 0.007) and fat mass to fat-free mass ratio (−6.6%, *p* = 0.031) were all lower at the end of the dietary intervention compared to baseline (Table 5). The average maternal BMI shifted from overweight during the baseline period to normal range after the dietary intervention phase. Infant weight, length and head circumference all increased significantly across the 2-week dietary intervention compared to baseline (Table 5). Infant rate of growth was stable across the study (week 1 ±SD vs. week 3 ± SD) according to weight-for-length z-score (0.10 ± 0.88 vs. 0.13 ± 0.96, *p* = 0.83), weight-for-age z-score (−0.17 ± 0.69 vs. −0.14 ± 0.63, *p* = 0.29) and length-for-age z-score (−0.27 ± 0.99 vs. −0.12 ± 0.56, *p* = 0.08).

### 3.6. Associations between HM Composition and Production at Baseline, Maternal Body Composition and Infant Growth and Changes in HM Metabolic Hormones and Dietary Intake

Univariate linear mixed-effects models were fit to determine if (a) HM composition and production were associated with maternal anthropometric measurements at baseline (as has been reported in some previous studies), (b) whether the changes in HM hormone concentrations were related to changes in maternal dietary intake, body weight or fat mass and (c) whether there was any evidence of an association between HM composition/production and infant growth. 

There was no relationship between any macronutrient and maternal anthropometric measurements or HM production (data not shown). There was a significant, positive association between HM leptin concentrations at baseline and maternal body weight (parameter estimate, *p* value: 0.01, *p* < 0.001), BMI (0.03, *p* < 0.001), fat mass (0.02, *p* < 0.001) and fat mass to fat-free mass ratio (0.98, *p* < 0.001) at this same timepoint. There was no association between any measures of maternal body composition and concentrations of HM adiponectin or insulin.

The change in HM leptin concentrations during the intervention was positively associated with the changes in maternal body weight (*p* < 0.001) and maternal fat mass (*p* < 0.001). No associations were identified between the change in HM adiponectin or insulin concentrations and changes in maternal fat mass or body weight during the study. In addition, the change in HM adiponectin concentrations was positively associated with the changes in maternal carbohydrate (*p* = 0.033) and total energy (*p* = 0.038) intake. There were no associations identified between the change in HM insulin or leptin concentrations and changes in maternal fat and sugar intake.

Infant length (0.005, *p* = 0.049) and head circumference (0.004, *p* = 0.022) across all study weeks were positively correlated with 24 h milk production (parameter estimate, *p* value). Additionally, only infant head circumference was positively correlated with the intake of fat consumed by the infant (0.092, *p* = 0.026) across all weeks. There was no association between any measures of infant growth and amount of protein or lactose consumed.

## 4. Discussion

The results of this study indicate that improving maternal dietary quality during lactation by lowering intakes of energy, fat, saturated fat, carbohydrate and sugar for as little as 2 weeks is associated with significant reductions in maternal body weight and fat mass, and in concentrations of the metabolic hormones leptin, insulin and adiponectin in HM, as well as infant intakes for leptin and insulin. The dietary intervention did not, however, affect either the concentrations of fat, protein or lactose in HM, 24 h milk production or infant growth rate over the same 2-week period.

This study is the first to directly assess the impact of a dietary intervention specifically aimed at reducing saturated fat and sugar intake in lactating women on the composition of their milk. The lack of significant changes in the concentrations of fat, protein or lactose in the current study is consistent with the results of the only previous study to assess the impact of a low-fat, high-carbohydrate diet in lactating women, which also found no changes in HM macronutrient composition across an 8-day period [8]. Interestingly, the previous study showed that consuming a high-fat diet, low-carbohydrate diet for 8 days resulted in increased fat and energy content in HM in lactating women between 6 and 14 weeks postpartum, although HM sampling did not include fat concentrations from a 24 h collection. This aligns with data from an animal study in which feeding rat dams a high-fat, high-sugar diet during lactation was associated with increased fat content of the milk, but no changes in the levels of protein [12]. A study by Yahvah and colleagues also reported that maternal intake of high-fat (36% of daily energy) compared to low-fat dairy (24% of daily energy) was associated with increased fat content in HM [32]. It is important to note, however, that this study was conducted at around 6 months postpartum, and in partially breastfeeding women who were feeding substantial amounts of formula and complementary foods to their infants. Thus, given the rapid changes in milk composition during weaning, interpreting these findings is difficult. 

The results of our study are aligned with a previous study conducted in exclusively or predominately breastfeeding women that suggests that reducing fat intake for a similar period is not associated with reductions in HM fat content [8]. The level of fat in HM is the primary determinant of its energy content [33], and it may be that the maintenance of HM fat content in response to decreased maternal fat intakes is a mechanism to maintain energy supply to the infant in the face of maternal nutrient restriction. This is supported by the results of studies in which women were subjected to intermittent fasting (e.g., during Ramadan) or periods of energy restriction, which indicate that fat content of HM is maintained even when maternal nutrient intake is significantly decreased [34,35,36,37].

In contrast to the macronutrients, HM concentrations of the metabolic hormones, leptin, insulin and adiponectin, were all decreased (by 10–25%) at the end of the dietary intervention compared to baseline. To our knowledge, no previous studies have directly investigated the effects of a maternal dietary intervention aimed at reducing maternal fat and sugar intakes during lactation on the concentrations of metabolic hormones in HM. Studies in lactating rats have reported that consumption of high-fat diets (8%, 25% and 60% of daily energy vs. 2%, 5% and 17% from the control diet, respectively) is associated with increased concentrations of leptin [38,39] and/or insulin [40] in the milk, but the effects of reducing fat intakes on HM metabolic hormone concentrations have not been reported. In humans, dietary counselling in pregnant women, aimed at achieving fat and fibre intake aligned with dietary recommendations, has been associated with increased concentrations of adiponectin in colostrum, but the effects on hormone concentrations in transitional and mature HM were not assessed [41].

Previous studies have shown that metabolic hormones in HM can be derived from both the synthesis in mammary epithelial cells and maternal circulation [42,43,44]. One possibility is that the reductions in metabolic hormones in HM in the current study were secondary to decrease in maternal circulating concentrations. This is supported by our finding that HM leptin concentrations were associated with the changes in maternal body weight and fat mass, although this did not appear to be the case for the other hormones. Furthermore, previous studies have reported that decreases in intake of energy and sugar, such as those in the current study, are associated with lower circulating glucose and insulin concentrations [45,46], both of which have been reported to be related to insulin concentrations in mature HM [47]. Similarly, HM adiponectin and leptin concentrations are directly related to the levels of these respective hormones in maternal circulation [48,49,50], and HM leptin has also been shown to be positively correlated to maternal body fat mass in both this and prior studies [51,52]. The relative contribution of reduced concentrations of metabolic hormones in maternal circulation and reduced synthesis within the mammary gland to the reduced levels of insulin, leptin and adiponectin at the end of the dietary intervention in this study is, however, unclear. The long-term consequences of these changes on the infant are also of interest, given that metabolic hormone concentrations in HM in the early postpartum period have been related to growth and fat deposition in the infant in the first few months after birth and to long-term risk of obesity [2,53].

Similar to HM macronutrient content, the findings of the current study suggest that a short-term reduction in energy and fat intake did not reduce 24 h milk production, which aligns with the results of the study described earlier, in which women consumed a low-fat, high-carbohydrate or high-fat, low-carbohydrate diet for 8 days [8]. As a result, the average amount of fat, protein and lactose supplied to the infant was not altered by the dietary intervention. This was reflected in the maintenance of infant growth across the dietary intervention period, and the positive association between 24 h milk production and infant growth supports the suggestion that HM milk intake is a critical driver of infant growth in infants who are exclusively breastfed [34]. Overall, these findings suggest that milk production, as well as HM macronutrient composition, is maintained in the face of a short-term reduction in maternal energy, fat and sugar intakes.

We are confident that compliance with the dietary intervention was high, as evidenced both by the dietary intake data collected from women before and during the intervention and the significant reductions in body weight (−1.8%) and body fat mass (−6.3%) that were observed over the study period. These results are in line with a systematic review which reported that dietary interventions were effective at reducing body weight and fat mass in lactating women [54]. The results of the present study add to this previous literature by suggesting that women can safely reduce energy intakes for a short term during lactation without negative effects on milk production, HM macronutrient content or infant growth. It is important to note, however, that the dietary intervention here was applied for just 2 weeks, and the effects of longer dietary interventions on HM composition and production remain to be determined. Further, a 2013 study conducted in Greece reported that lactating women exclusively breastfeeding showed a significant weight loss of 0.7 kg per month in the first trimester and a nonsignificant weight loss of 0.5 kg per month in the second trimester [55]. Our study showed that lactating women were able to lose an average of 1.3 kg during a 2-week period, which suggests that the dietary intervention was able to produce greater weight losses than those that would be expected based on the increased energy demands of lactation alone.

A major strength of this study is the use of a home delivery service for the delivery of meals and snacks during the dietary intervention, which ensured a high degree of compliance with the dietary intervention. Another strength is the extensive sampling regimen, with daily collection of samples across the study period and from both breasts, in order that any rapid shifts in composition could be monitored. In addition, no women in the study sample had health conditions, such as diabetes and gestational diabetes, or were smokers, all of which have been shown to impact HM composition and could therefore have confounded the results [56,57]. 

While the study had many strengths, there were also limitations, the main ones being the small sample size and relatively short duration of the dietary intervention; thus, it is not possible to draw definitive conclusions as to the effect of longer durations of reduced fat and sugar intakes on HM composition, and further studies are needed. It is also important to note that our study population comprised predominately Caucasian women of a higher socioeconomic status, and that the results may not be representative of the outcomes that would be obtained if a similar dietary intervention were applied in different ethnic populations from a diverse educational and socioeconomic background.

## 5. Conclusions

In conclusion, the results of the study suggest that improving maternal dietary quality during lactation by increasing fibre intake and reducing energy, fat, carbohydrate and sugar intakes for a 2-week period significantly reduced both maternal body weight and fat mass and reduced concentrations of leptin, insulin and adiponectin in HM. HM macronutrient concentrations, milk production and infant growth were unaffected. While further studies are required to assess the impact of longer durations of maternal dietary interventions during lactation and their impact on infant short- and long-term health, these results suggest that there are no adverse impacts of a short duration of modest maternal energy restriction on milk production or macronutrient content. These findings should provide reassurance to women who seek to lose body weight and fat mass while breastfeeding, although further studies are required to assess any potential effects of the reduced leptin, adiponectin and insulin concentrations in HM on the growth and/or adiposity of the infant.

## Figures and Tables

**Figure 1 nutrients-13-01892-f001:**
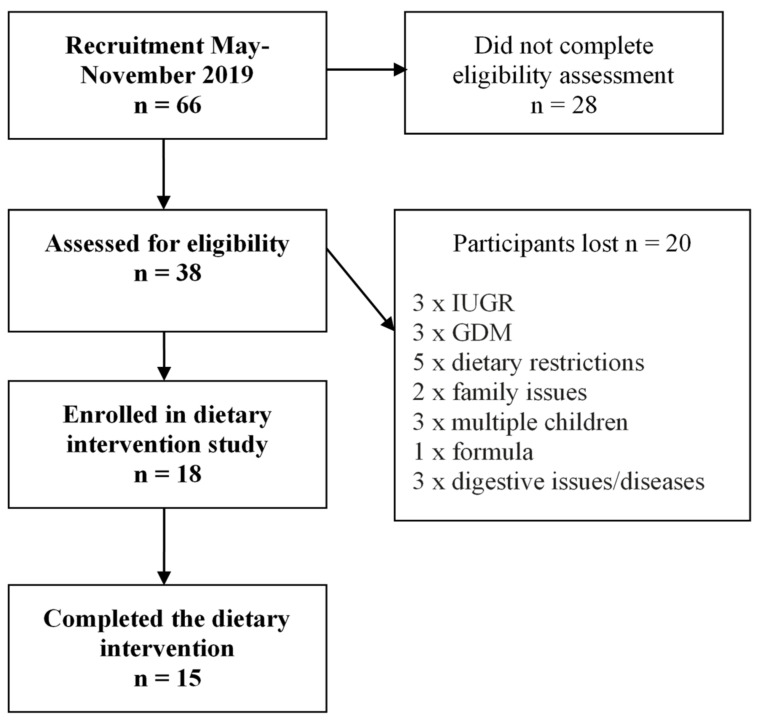
Flow chart of participants from recruitment to completion of study ^1^. Three participants did not complete all three weeks of sample collection (two withdrew during week 1 of the study and one only completed the first 2 weeks). ^1^ GDM, gestational diabetes; IUGR, Intrauterine growth restriction.

**Table 1 nutrients-13-01892-t001:** Summary of sample collection for analysis of human milk composition across the entire 3 weeks of the study ^1^.

Sample Type	Frequency of Collection	Time of Collection	Procedure	Total N
Daily	Once a day (1 sample per breast)	Morning before the first morning feed	Prefeed samples from both breasts	42 (21 samples per breast)
Intensive sampling	Once a week (2 samples per breast)	One in the afternoon and one in the evening in addition to daily sample	Prefeed samples from both breasts	12 (6 samples per breast)

^1^ N, total number of samples by completion of 3 weeks.

**Table 2 nutrients-13-01892-t002:** Baseline clinical characteristics of participating mothers and infants at start of dietary intervention study ^1^.

Characteristics	Mean ± SD	Range
**Mothers**		
Age (years)	32 ± 3	27–37
Pre-pregnancy weight ^2^ (kg)	69.4 ± 10.8	55–89
Pre-pregnancy BMI ^2^ (kg/m^2^)	25.1 ± 4.1	17.2–32.7
Current weight (kg)	71.4 ± 10.3	54.4–80.7
Current BMI (kg/m^2^)	25.8 ± 4.1	17.1–33.0
**Infants**		
Age (months)	3.1 ± 0.8	1.6–4.9
Sex ^2^ (M/F)	8/6	-
Birth weight ^2^ (kg)	3.6 ± 0.4	2.9–4.3
Birth length ^2^ (cm)	51.1 ± 1.7	48–53
Current weight (kg)	6.1 ± 1.0	4.6–7.6
Current length (cm)	60.3 ± 3.3	55.2–65.5

^1^ Values are shown as mean ± standard deviation. BMI, body mass index; F, female; M, male, ^2^ Missing information from one participant.

**Table 3 nutrients-13-01892-t003:** Overview of daily intake of key nutrients before and during the dietary intervention phase ^1^.

	Week 1 Baseline ^2^	Week 2 Intervention	Week 3 Intervention	Intervention Phase Combined
Energy (kcal)	2525 ± 579	1716 ± 84 *	1678 ± 134 *	1697 ± 104 *
Protein (%en)	17 ± 3	22 ± 1 *	22 ± 1 *	22 ± 1 *
Protein (g)	105 ± 24	93 ± 8 *	91 ± 8 *	92 ± 7 *
Carbohydrate (%en)	40 ± 5	47 ± 2 *	47 ± 2 *	47 ± 2 *
Carbohydrate (g)	254 ± 62	202 ± 12 *	196 ± 15 *	199 ± 12 *
Fibre (g)	29 ± 7	34 ± 3 *	32 ± 3	33 ± 3 *
Total sugars (g)	116 ± 45	83 ± 5 *	80 ± 8 *	82 ± 5 *
Fat (%en)	40 ± 6	27 ± 2 *	28 ± 2 *	28 ± 2 *
Fat (g)	114 ± 38	52 ± 5 *	52 ± 7 *	52 ± 6 *
Saturated fat (%en)	14 ± 9	9 ± 1 *	9 ± 1 *	9 ± 1 *
Saturated fat (g)	46 ± 17	17 ± 2 *	17 ± 3 *	17 ± 2 *

^1^ Values are shown as mean ± standard deviation; daily intake for key nutrients shows values for Week 1, Week 2, Week 3 and all 14 days of the intervention combined (intervention phase combined). Asterisks (*) denote values that were significantly different from values at baseline (*p* < 0.05) (Week 1). There were no significant differences between any values in Week 2 and Week 3 of the intervention. All *p*-values were computed using univariate linear mixed-effects models with individual mothers fitted as random effects. Percentage of energy (%en) is provided for protein, carbohydrate, fat and saturated fat. Carbohydrate is defined as a sum of total sugars, maltotriose, starch and other available carbohydrates (glycogen + raffinose + stachyose + dextrins + maltodextrins + other undifferentiated oligosaccharides). Total sugar is defined as a sum of fructose, glucose, sucrose, maltose, lactose and galactose according to software FoodWorks 10 Professional and Australian Food Composition Database. ^2^ Missing information from one participant.

**Table 4 nutrients-13-01892-t004:** Overview of human milk macronutrient and metabolic hormone concentrations before and after dietary intervention ^1^.

Macronutrient and Metabolic Hormones	Baseline Period	Intervention Phase		*p* Value
	Week 1	Week 2	Week 3	
Fat (g/L)	34 ± 13	34 ± 16	35 ± 14	0.63
Protein (g/L)	8 ± 3	8 ± 2	8 ± 3	0.30
Lactose (g/L)	65 ± 22	61 ± 20	63 ± 23	0.75
Leptin (ng/mL)	0.25 ± 0.17	0.23 ± 0.18	0.20 ± 0.17 *	**<0.0001**
Insulin (µIU/mL)	28.70 ± 20.79	18.91 ± 12.28 *	21.49 ± 13.88 *	**<0.0001**
Adiponectin (ng/mL)	12.63 ± 6.44	11.80 ± 5.24	11.35 ± 5.56 *	**0.048**

^1^ Values are shown as mean ± standard deviation. Asterisks (*) denote values that were significantly different from values at baseline (*p* < 0.05) (Week 1). *p*-values in bold are significant (*p* < 0.05) and represent comparisons between the intervention phase (Week 2 or Week 3) and the baseline period (Week 1). All *p*-values were computed using univariate linear mixed-effects models with individual mothers fitted as random effects.

**Table 5 nutrients-13-01892-t005:** Maternal body composition and infant growth measurements across the study ^1^.

Anthropometry and Body Composition	Baseline Period	Intervention Phase		PE ± SE ^1^	*p*
	Week 1	Week 2 ^2^	Week 3		
Mothers					
Weight (kg)	71.4 ± 10.3	70.6 ± 10.3	70.1 ± 9.8 *	−1.39 ± 0.35	**0.002**
BMI (kg/m^2^)	25.8 ± 4.1	25.6 ± 4.1	24.3 ± 4.0 *	−0.50 ± 0.13	**0.002**
Fat-free mass (kg)	44.2 ± 4.7	45.2 ± 4.6	44.5 ± 4.0	0.32 ± 0.58	0.60
Fat mass (kg)	27.2 ± 7.1	25.4 ± 6.7	25.5 ± 6.5 *	−1.70 ± 0.54	**0.007**
Fat mass (%)	37.6 ± 5.4	35.5 ± 5.0	35.9 ± 4.9	-	-
Fat mass to fat-free mass ratio	0.61 ± 0.1	0.56 ± 0.1	0.57 ± 0.1 *	−0.05 ± 0.02	**0.031**
Infants					
Weight (kg)	6.1 ± 1.0	6.3 ± 1.0 *	6.5 ± 1.0 *	0.43 ± 0.05	**<0.001**
Length (cm)	60.3 ± 3.3	61.1 ± 3.0 *	61.9 ± 3.1 *	1.53 ± 0.15	**<0.001**
Head circumference (cm)	40.4 ± 2.0	40.8 ± 1.7 *	41.4 ± 1.6 *	0.99 ± 0.14	**<0.001**

^1^ Effects of predictors taken from univariate linear mixed-effects models comparing baseline period (Week 1) to end of intervention phase (Week 3). PE, parameter estimate; SE, standard error. Asterisks (*) denote values that were significantly different from values at baseline (*p* < 0.05) (Week 1). *p*-values in bold are significant (*p* < 0.05) and represent comparisons between the intervention phase (Week 2 or Week 3) and the baseline period (Week 1). All *p*-values were computed using univariate linear mixed-effects models with individual mothers fitted as random effects. ^2^ Missing information from one mother and her infant.

## Data Availability

The data presented in this study are available on request from the corresponding author. The data are not publicly available for privacy reasons.

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
