# Peer review of "Reduction in Maternal Energy Intake during Lactation Decreased Maternal Body Weight and Concentrations of Leptin, Insulin and Adiponectin in Human Milk without Affecting Milk Production, Milk Macronutrient Composition or Infant Growth"

_nutrients, 2021, doi:10.3390/nu13061892_

Round 1

Reviewer 1 Report

This study investigates the effects of improving maternal diet during lactation (decrease in energy intake and improvement of nutrient quality) on several parameters of human milk (HM) (production, nutrients and hormones concentrations) that could be relevant for infant growth and development. Maternal and infant physiology are also reported. This is a relevant and well-written study in the field of breastfeeding in order to better understand the relationship between maternal nutrition, breast milk composition and infant growth. However, several aspects of the study need clarifications.

  1. Dietary intervention.

- Could you justify/explain why you choose a 2-week intervention (and not shorter or longer length)?

- line 100: Could you clarify: “…three 24-h dietary recalls (2 week? and 1 weekend day)…”

- Could you clearly define what you consider as (added) sugar and carbohydrate?

- line 118: Could you provide the number of women included in the study that have food allergies or intolerances?

- can you precise what you consider “carbohydrate” and “sugar”?

  1. Measurements and collection

- I assumed that no maternal blood samples were collected during the study?

- line 144: “all measurements”, which measurements exactly were performed in duplicate? Why after a breastfeeding session? Could you clarify.

- The protocol of HM collection requires clarifications. Line 155: “collect HM samples each morning…” For the all duration of the study? ie 3 weeks (21 days?). What do you call “intensive milk sampling once a week throughout the study”? Line 178: “Milk samples were collected before and after every feed”. Overall, it would be very helpful to provide a table or figure explaining/summarizing the multiple HM samples (time of collection, procedure etc.) and what they were used for exactly (line 277-281…). Did all women collect HM samples daily?

- Can you provide the number of women performing hand expression or pump collection?

- Line 200: what are “the standard assays”? Clarify the text.

- Metabolic hormone measurements: what dilution of HM do you use in each assay?...as whole HM can contain inhibiting factors for ELISA.

  1. Results.

- Table 2 present the energy intake between baseline and the intervention phase. Was there any difference during week 1 and week 2 of intervention? Or was the reduction in energy intake obtained since week 1 of intervention? It would be useful to have this information since you present HM factor concentrations per week in Table 3.

- Amount of protein, carbohydrate etc. are presented as %en but also as g. Could you thus indicate the daily intake in g as well?

- Table 3: It is not clear what comparison you make when you indicate “a” and “b”? Please, specify. Also, indicate in the legend of all tables the statistical model used.

- Paragraph 3.6. It is not clear what associations you’re testing here. Why only at baseline? Could you give more rationale? Also, are there any correlation between change in HM metabolic hormones and any maternal anthropometric measurements during the interventions? It would be interesting to test what explain best these significant changes?

- Shouldn’t the title of the study also reflect your data on HM metabolic hormones? These results are emphasized in several part of the article (e.g. first sentence of the conclusions)

- Paragraph 3.4. Could you also calculate and include the infant’s 24h intakes of HM insulin, leptin and adiponectin at each week? As HM hormone concentrations varies between week1 and week3, it would be expected to have an impact on infant’s intake?

Author Response

17 May 2021

Cover letter to editor and overview of the most important revisions made

Dear Ms. Hulda Chen,

We are pleased that the editor and reviewers felt that our manuscript has merit and thank them for their careful and thorough review of the manuscript. We have worked to address the comments raised in the revised version of the manuscript and provided point-by-point responses to the reviewer comments below. For ease of identification, we have included line and page references for all amendments to the manuscript and also highlighted revisions in yellow in the revised document.

We have paid particular attention to the comments from the reviewers related to the methods section, and added further details related to the study hypothesis, dietary intervention, milk sampling, 24-h milk production, sample preparation and statistical analysis, in order to improve clarity. In addition, we have also worked to perform additional analyses to address infants’ 24-h intakes for metabolic hormones across the study and to determine whether changes in hormone concentrations in HM were associated with changes in either maternal body weight/fat mass and/or changes in intake of specific nutrients.

We thank the reviewers for their valuable suggestions. We feel that the manuscript has benefited from the Editorial process and look forward to hearing whether it is now acceptable for publication in Nutrients.

Sincerely,

A/Prof Bev Muhlhausler

Author's Reply to the Review Report (Reviewer 1)

 Comments and Suggestions for Authors

This study investigates the effects of improving maternal diet during lactation (decrease in energy intake and improvement of nutrient quality) on several parameters of human milk (HM) (production, nutrients and hormones concentrations) that could be relevant for infant growth and development. Maternal and infant physiology are also reported. This is a relevant and well-written study in the field of breastfeeding in order to better understand the relationship between maternal nutrition, breast milk composition and infant growth. However, several aspects of the study need clarifications.

  1. Dietary intervention.

- Could you justify/explain why you choose a 2-week intervention (and not shorter or longer length)?

As this was a proof-of-concept study, we estimated the timeframe required to identify changes in human milk composition based on limited previous literature and opted for an intermediate duration for the intervention. We reasoned that too short a period (i.e. a few days) would be insufficient for us to draw any clear conclusions, whereas if changes in composition were not observed within 2 weeks there was unlikely to be a rationale for a longer study. The rationale for the choice of dietary intervention timeframe has been added to the revised manuscript [Page 2, Lines 93-95].

- line 100: Could you clarify: “…three 24-h dietary recalls (2 week? and 1 weekend day)…”

This is intended to indicate that of the three 24-h dietary recalls, two were conducted on weekdays and one was conducted on a weekend day. This information has been added to the revised manuscript [Page 3, Lines 110-111].

- Could you clearly define what you consider as (added) sugar and carbohydrate?

We have clarified that the aim of the dietary intervention was to limit intake of total sugars by targeting added sugars and a statement to this effect has now been included [Page 2, Lines 90-93]. Total sugar intake was defined as a sum of fructose, glucose, sucrose, maltose, lactose and galactose, while carbohydrate was defined as a sum of total sugars, maltotriose, starch and other available carbohydrates (glycogen + raffinose + stachyose + dextrins + maltodextrins + other undifferentiated oligosaccharides). These values were determined using the software FoodWorks 10 Professional and Australian Food Composition Database. This information is now included in the text and description of Table 3 in the revised manuscript [Page 9, Lines 305-308].

- line 118: Could you provide the number of women included in the study that have food allergies or intolerances?

This information was captured in the baseline questionnaire and none of the women in the study reported having any food allergies or intolerances [Page 3, Lines 129-130].

  1. Measurements and collection

- I assumed that no maternal blood samples were collected during the study?

We did not collect maternal blood samples in this study. Since this was a proof-of-concept study and we were already performing a large number of measurements and human milk collections, we opted to reduce participant burden by focusing only on components in the milk. However, we agree that the ability to measure hormones in maternal blood would have added additional information, particularly in relation to the mechanisms underlying the shift in HM metabolic hormone concentrations and would recommend inclusion of this parameter in future studies.

- line 144: “all measurements”, which measurements exactly were performed in duplicate? Why after a breastfeeding session? Could you clarify.

Measurements for fat mass, percentage fat mass and fat-free mass were performed in duplicate. As per standard practice, maternal body composition was collected after a breastfeeding session in order to avoid any interference, including the variable milk volume of each breastfeeding session. These details have been added to the methods section of the revised manuscript [Page 4, Line 158-160].

- The protocol of HM collection requires clarifications. Line 155: “collect HM samples each morning…” For the all duration of the study? ie 3 weeks (21 days?). What do you call “intensive milk sampling once a week throughout the study”? Line 178: “Milk samples were collected before and after every feed”. Overall, it would be very helpful to provide a table or figure explaining/summarizing the multiple HM samples (time of collection, procedure etc.) and what they were used for exactly (line 277-281…). Did all women collect HM samples daily?

A pre-feed breast milk sample was collected every morning throughout the study. In addition to this – on one day of each study week, women undertook ‘intensive HM sampling’ which involved collecting a pre-feed milk sample at 3 separate time-points (morning, afternoon and evening) over a 24-hour period [Page 4, Lines 168-170]. We have included an additional table (new Table 1) in the revised manuscript which includes further details of the sample collection protocol [Page 4, Lines 174-175; Page 5, Lines 182-183].

Four women did not complete collection of all daily and intensive samples for both breasts, and this information has been added to the results section of the manuscript: As per the study protocol, the total number of HM samples collected across the study period for each woman was 54 samples (27 from each breast) [Page 4, Lines 173-174]. ‘Eleven women provided a complete set of 54 HM samples and the remaining 4 women provided between 47 and 53 HM samples’ [Page 8, Lines 287-288].

- Can you provide the number of women performing hand expression or pump collection?

Women were allowed to select their preferred method of milk collection. We did not capture data on the number of women selecting hand expression vs pump collection, since in our previous experience the HM composition does not differ between these methods of collection.

- Line 200: what are “the standard assays”? Clarify the text.

We have amended this statement to: The assays for macronutrient and metabolic hormone measurements (all previously validated for use in human milk), since we agree that ‘standard assays’ was not entirely clear [Page 6, Lines 218-220].

- Metabolic hormone measurements: what dilution of HM do you use in each assay?...as whole HM can contain inhibiting factors for ELISA

The dilution of HM used was different for each assay, and information on the dilution used is now included in the text [Page 6, Lines 238-239, 243, 248].For the adiponectin assay, the samples were diluted 1:3 in buffer, whereas undiluted HM was used for the insulin and leptin measures. All kits have been extensively validated and optimised and used in a large number of previous studies in our laboratory, and we are confident that the HM matrix did not materially affect the performance of the ELISA.

  1. Results.

- Table 2 present the energy intake between baseline and the intervention phase. Was there any difference during week 1 and week 2 of intervention? Or was the reduction in energy intake obtained since week 1 of intervention? It would be useful to have this information since you present HM factor concentrations per week in Table 3.

Thank you for raising this point. Table 3 (former Table 2) has been updated and now includes values for each of the weeks of the intervention as well as combined values for the whole intervention period. Weeks 2 and 3 (intervention) showed the same statistical significance as data from the intervention phase combined, except for fibre intake on week 3, and there were no differences between week 2 and week 3 of the intervention. Table legend has also been updated [Pages 8-9, Lines 301-308].

- Amount of protein, carbohydrate etc. are presented as %en but also as g. Could you thus indicate the daily intake in g as well?

We have presented the intake of different macronutrients as both g and % energy, so are not sure what information the reviewer wishes us to add. We do not consider that presenting total daily intake in grams would not add any meaningful information, and that g intake of each component in addition to total energy consumed is more helpful.

- Table 3: It is not clear what comparison you make when you indicate “a” and “b”? Please, specify.

We have replaced the superscripts with asterisks to indicate values that are significantly different from baseline (Week 1) according to the univariate model. The figure legend has been updated to clarify this point [Page 9, Lines 315-320].

- Also, indicate in the legend of all tables the statistical model used.

All tables have been updated to include explanations of the statistical models used [Page 9, Lines 304-305, Lines 319-320; Page 10, Lines 348-349].

- Paragraph 3.6. It is not clear what associations you’re testing here. Why only at baseline? Could you give more rationale? Also, are there any correlation between change in HM metabolic hormones and any maternal anthropometric measurements during the interventions? It would be interesting to test what explain best these significant changes?

Thank you for this comment. These were exploratory analyses with which we aimed to determine whether there was any evidence that 1) macronutrient and hormone concentrations were associated with maternal anthropometric measurements and HM production at baseline, and 2) if 24-h milk production and doses of fat, protein and lactose were associated with infant anthropometric measurements. The comparisons were only performed with values collected at baseline, since the primary objective was to determine whether (a) there was any evidence that HM composition was related to maternal anthropometric measures at the start of the study (as this has been reported in other studies) and (b) whether there was evidence that HM composition before the intervention was associated with infant growth across the study period.

We have also conducted additional analyses to explore whether there were relationships between changes in maternal body composition/body weight and changes in hormone concentrations. The results of these analyses suggest that the change in leptin concentrations was positively associated with the change in maternal body weight (p<0.001) and maternal fat mass (p<0.001), but there were no relationships between changes in either maternal body weight or fat mass and changes in levels of other hormones in HM. In addition, the change in HM adiponectin concentrations was positively associated with the changes in maternal carbohydrate (p=0.033) and total energy (p=0.038) intake. There were no associations identified between the change in HM insulin or leptin concentrations and changes in maternal fat and sugar intake. This information has now been added to the revised manuscript [Page 10, Lines 367-374]. While these data may suggest that the changes in HM adiponectin are more directly related to the change in diet composition, whereas changes in HM leptin are more related to the associated shift in maternal body weight/fat mass, we acknowledge that given the small sample size it is not possible to draw any clear conclusions and further studies are needed.

The paragraph 3.6 has been updated to include clarifications and further analyses [Page 10, Lines 351-358, Lines 367-374]. Also, a sentence in the discussion has been included [Pages 11-12, Lines 433-435]

- Shouldn’t the title of the study also reflect your data on HM metabolic hormones? These results are emphasized in several part of the article (e.g. first sentence of the conclusions)

We thank the reviewer for this suggestion and agree that the title should ideally contain information about the major finding of the study. We have modified the title based on this suggestion to: “Reduction of maternal energy intake during lactation decreased maternal body weight and concentrations of leptin, insulin and adiponectin in human milk without affecting milk production, milk macronutrient composition, or infant growth”.

- Paragraph 3.4. Could you also calculate and include the infant’s 24h intakes of HM insulin, leptin and adiponectin at each week? As HM hormone concentrations varies between week1 and week3, it would be expected to have an impact on infant’s intake?

Thank you for raising this point. We have calculated the infant’s intake for each of the metabolic hormones and identified that intake of insulin and leptin were also reduced in Week 3 compared to Week 1. Please note that 24-h milk production was measured at baseline and at the end of the intervention phase, and therefore, values for milk intake are not available for Week 2. Paragraph 3.4. has now been updated to include these new data [Page 9, Lines 325-326, 330-332] and a sentence included in the revised discussion section [Page 11, Line 387].

Reviewer 2 Report

Major concerns:

  • There is not a hypothesis presented
  • The age range for recruitment spanned 6 – 20 weeks, and the age range of infants at the time of participation was 1.6 – 4.9 months. This large range likely significantly confounded the results. They should provide evidence that milk components were not different just based on lactation stage/time (e.g. a control group that is age matched without the intervention) or adjust the models for age of the child
  • The authors did not measure “milk production”, rather they measured infant milk intake. It is suggested to remove the “milk production” conclusions and present these data as intake
  • HEI score should be calculated as a measure of diet quality and used in models as an explanatory variable rather than “week”
  • Maternal serum parameters should be presented to aid in the understanding of how the diet intervention impacted maternal circulating hormones. This may also help to explain the changes observed in HM composition

Detailed Comments:

Abstract

Line 39 – 40 – the statement that: “changes in HM hormones have the potential to influence infant growth and adiposity” is a little too strong and not very well justified. Either soften the conclusions or provide background that suggests these hormones can influence infant growth and adiposity as your data indicate no change in infant growth.

Introduction

  • Line 51: Please provide a reference for this statement as I do not believe the general consensus is that maternal diet significantly impacts HM composition. In fact, the sentence on lines 55-56 contradict the statement. You may be better off introducing the topic by stating that very little direct evidence exists to suggest that maternal diet can impact HM composition.
  • You might want to consider using the word “lactation” rather than “breastfeeding” where applicable throughout the introduction and manuscript (example line 74). Lactation is the physiological state, whereas breastfeeding is the act of feeding a child from the breast. A lactating woman can still produce HM without actually breastfeeding.
  • The introduction contains a clear study objective (lines 75-79), however there is not a hypothesis presented. Please include what was hypothesized to happen with this dietary intervention

Methods

  • 6 – 20 weeks is a huge range in age for this kind of study. Milk composition is largely dependent on lactation stage/postnatal time and therefore it is likely to significantly confound these results.
  • Why was a healthy eating index score not used to assess the diet quality pre and post intervention?
  • Line 170 – 179 “24-h Milk Production”: the authors describe a protocol used for 24h milk production and report on these results as such. However, the procedure that they describe does not sufficiently estimate milk production. Rather it is a measure of infant intake, which is quite different that the actual production of milk by the mammary gland. For example, if this procedure accurately measured milk production, the participant should not have been able to collect any more milk after the feed. The authors should not present these data as 24h milk production and only report the infant intakes of HM over the 24h period. As it is currently presented, the data are not correct and misleading
  • Line 180: the use of the word “Dose” is not appropriate. Please use “intake” or something similar instead
  • Line 208: please provide the equation used
  • Line 247: “data were” not “data was”
  • Line 252: “analyses” not “analysis”
  • Line 253: “were” not “was”
  • Statistics: in your models you present “week” as an explanatory variable which should indicate the effect of the intervention on your outcomes. However, the use of a diet quality measure such as HEI rather than “week” would provide much more insight into the potential relationship between maternal diet and HM composition
  • Statistics: Please describe what descriptive statistics are presented (eg means SD or median IQR and did you use t-test or non-parametric tests)

Results

  • Table 1: Maternal pre-pregnancy BMI and postnatal BMI was not different, suggesting that these women lost all of their pregnancy weight by the time they participated in the study. It not clear why this is the cohort chosen for a calorie-restriction/weight-loss intervention and would seem to be more clinically relevant and important in a population of women who had a BMI that indicated overweight and/or obesity. Please discuss the generalizabity and clinical importance of your findings considering the maternal BMI of this cohort.
  • Table 2: Again it would be nice to see the change in dietary quality (e.g. HEI score)
  • Table 2: What statistical test was used? Where all data normally distributed?
  • Compliance to the dietary intervention should be assessed and reported in either the methods or the results
  • Table 3: What statistical test was used?
  • Line 307 – 308: milk production was not measured – infant volume intake of HM was measured
  • Table 4: None of the models are discussed in the results section. Only the pre-post changes to maternal anthros and body comp. Please explain/present the models
  • You present data on the relationship between HM leptin and maternal body comp, but none of the other HM components? Also you do not mention if there was any associations at timepoints other than at baseline… Please present these data.
  • Also it is necessary that you model HM components with diet quality or the change in dietary components XYZ as a predictor in order to actually accomplish your aim of “determine the effect of reducing energy, fat and sugar intakes in lactating women over a 2-week period on the concentrations of macronutrients and metabolic hormones in HM”. As currently presented you have not actually completed your primary aim.

  • Line 333: milk production – again you did not measure this, you measured infant intake of HM
  • Line 334: use of the word “dose” please change to “amount”
  • Was infant length correlated with fat intake across all weeks or just at one timepoint?

Discussion

  • The study would be significantly strengthened if you could provide maternal circulating levels of hormones, as discussed Lines 384 -399
  • There needs to be some rationale for doing such a short intervention. Do you think this will have any positive long-term outcomes for the moms or will they go back to eating the way they were before?
  • The statement 417 -419 should be softened or edited to say that a “short-term reduction in energy intake” will not affect HM production or infant growth.

Author Response

17 May 2021

Cover letter to editor and overview of the most important revisions made

Dear Ms. Hulda Chen,

We are pleased that the editor and reviewers felt that our manuscript has merit and thank them for their careful and thorough review of the manuscript. We have worked to address the comments raised in the revised version of the manuscript and provided point-by-point responses to the reviewer comments below. For ease of identification, we have included line and page references for all amendments to the manuscript and also highlighted revisions in yellow in the revised document.

We have paid particular attention to the comments from the reviewers related to the methods section, and added further details related to the study hypothesis, dietary intervention, milk sampling, 24-h milk production, sample preparation and statistical analysis, in order to improve clarity. In addition, we have also worked to perform additional analyses to address infants’ 24-h intakes for metabolic hormones across the study and to determine whether changes in hormone concentrations in HM were associated with changes in either maternal body weight/fat mass and/or changes in intake of specific nutrients.

We thank the reviewers for their valuable suggestions. We feel that the manuscript has benefited from the Editorial process and look forward to hearing whether it is now acceptable for publication in Nutrients.

Sincerely,

A/Prof Bev Muhlhausler

Author's Reply to the Review Report (Reviewer 2)

Comments and Suggestions for Authors

Major concerns:

  • There is not a hypothesis presented

The hypothesis for this proof-of-concept study was that improving maternal dietary quality would be associated with alterations in the concentrations of macronutrients and/or metabolic hormones in HM, in the absence of any effects on HM production. This has now been articulated in the introduction of the revised manuscript [Page 2, Lines 82-84].

  • The age range for recruitment spanned 6 – 20 weeks, and the age range of infants at the time of participation was 1.6 – 4.9 months. This large range likely significantly confounded the results. They should provide evidence that milk components were not different just based on lactation stage/time (e.g. a control group that is age matched without the intervention) or adjust the models for age of the child

We deliberately chose to include only women with established lactation (~1.5 and 4 months postpartum) who were exclusively breastfeeding a singleton infant in order to minimise the impact of changes in HM composition due to variations across lactation or substantive changes in infant feeding behaviour (particularly the introduction of solids). We are aware that HM composition varies across lactation, however, we would not anticipate significant changes in HM macronutrient content across a 2 week period in women in established lactation who are exclusively breastfeeding (Khan et al 2013), and therefore consider it highly unlikely that the changes observed were due to time effects.

Reference:

Khan S, Prime DK, Hepworth AR, Lai CT, Trengove NJ, Hartmann PE. Investigation of short-term variations in term breast milk composition during repeated breast expression sessions. Journal of Human Lactation 2013, 29(2), 196-204

The authors did not measure “milk production”, rather they measured infant milk intake. It is suggested to remove the “milk production” conclusions and present these data as intake

We assessed both milk production and milk intake in this study, using well-established methods that we have used in numerous previous studies by our group and others (see references below). Infant milk intake was assessed using the test weighing procedure (weighing the infant before and after each feed over a 24-h period and calculating volume of milk consumed at each feed) with the addition of any expressed milk fed to the infant. Milk production also included any milk that was expressed by the mother, but not fed to the infant, and was calculated using a validated equation () (Arthur et al 1987, Kent et al 2006). This approach has been shown to be equivalent to the dose to mother deuterium oxide method for measuring milk production (Butte 1988).We therefore consider that it is important to retain the distinction between milk intake and milk production in the revised manuscript.

References:

Arthur, P.; Hartmann, P.; Smith, M. Measurement of the milk intake of breast-fed infants. Journal of Pediatric Gastroenterology and Nutrition 1987, 6, 758-763.

Butte NF, Wong WW, Patterson BW, Garza C, Klein PD. Human-milk intake measured by administration of deuterium oxide to the mother: a comparison with the test-weighing technique. The American journal of clinical nutrition 1988; 47(5), 815-21.

Kent, J.C.; Mitoulas, L.R.; Cregan, M.D.; Ramsay, D.T.; Doherty, D.A.; Hartmann, P.E. Volume and frequency of breastfeedings and fat content of breast milk throughout the day. Pediatrics 2006, 117, 387-395.

George, A.D.; Gay, M.C.; Murray, K.; Muhlhausler, B.S.; Wlodek, M.E.; Geddes, D.T. Human milk sampling protocols affect estimation of infant lipid intake. The Journal of nutrition 2020, 150, 2924-2930.

  • HEI score should be calculated as a measure of diet quality and used in models as an explanatory variable rather than “week”

Thank you for raising this point. While we had considered using the HEI score, the central aim of our 2-week dietary intervention was to reduce intakes of discretionary foods, saturated fats and added sugars. Therefore, we opted to focus on the changes in intakes of these specific nutrients in this initial proof-of-concept study. While the HEI was beyond the scope of this initial study, we do agree that it is a valuable tool to assess compliance with dietary guidelines and would be valuable to include in future studies that further investigate the role of maternal diet (and changes in dietary quality) on HM composition.

  • Maternal serum parameters should be presented to aid in the understanding of how the diet intervention impacted maternal circulating hormones. This may also help to explain the changes observed in HM composition

Maternal blood samples were not collected in the study. As this was a proof-of-concept study and we were already performing a large number of measurements and human milk collections, we opted to reduce participant burden by focusing only on components in the milk. However, we agree that collection of maternal hormonal serum samples would have contributed to the study questions and recommend inclusion of this parameter in future studies.

Detailed Comments:

Abstract

Line 39 – 40 – the statement that: “changes in HM hormones have the potential to influence infant growth and adiposity” is a little too strong and not very well justified. Either soften the conclusions or provide background that suggests these hormones can influence infant growth and adiposity as your data indicate no change in infant growth.

We acknowledge that the current evidence base related to the relationship between HM hormone concentrations and infant growth and adiposity is not particularly strong, and have modified this sentence accordingly [Page 1, Lines 41-44].

“The limited studies to date that have investigated the association between metabolic hormone concentrations in HM and infant growth raise the possibility that the changes in HM composition observed in the current study could impact infant growth and fat deposition , but further studies are required to confirm this hypothesis.”

Introduction

  • Line 51: Please provide a reference for this statement as I do not believe the general consensus is that maternal diet significantly impacts HM composition. In fact, the sentence on lines 55-56 contradict the statement. You may be better off introducing the topic by stating that very little direct evidence exists to suggest that maternal diet can impact HM composition.

This sentence has been revised as suggested [Page 2, Lines 54-56], and now reads:

“One factor that has the potential to impact HM composition is variations in maternal diet, however very few studies have directly assessed the effect of maternal diets on HM composition”.

  • You might want to consider using the word “lactation” rather than “breastfeeding” where applicable throughout the introduction and manuscript (example line 74). Lactation is the physiological state, whereas breastfeeding is the act of feeding a child from the breast. A lactating woman can still produce HM without actually breastfeeding.

We have used the term ‘lactation’ to replace ‘breastfeeding’ throughout the manuscript where appropriate [Page 2, Lines 61, 77].

  • The introduction contains a clear study objective (lines 75-79), however there is not a hypothesis presented. Please include what was hypothesized to happen with this dietary intervention

We have now included the hypothesis for this study at the end of the introduction section of the revised manuscript [Page 2, Lines 82-84]:

“We hypothesized that the dietary intervention would result in significant changes in HM composition in the absence of any change in milk production or infant intake.”

Methods

  • 6 – 20 weeks is a huge range in age for this kind of study. Milk composition is largely dependent on lactation stage/postnatal time and therefore it is likely to significantly confound these results.

Please see response above.

  • Why was a healthy eating index score not used to assess the diet quality pre and post intervention?

Please see response above.

  • Line 170 – 179 “24-h Milk Production”: the authors describe a protocol used for 24h milk production and report on these results as such. However, the procedure that they describe does not sufficiently estimate milk production. Rather it is a measure of infant intake, which is quite different that the actual production of milk by the mammary gland. For example, if this procedure accurately measured milk production, the participant should not have been able to collect any more milk after the feed. The authors should not present these data as 24h milk production and only report the infant intakes of HM over the 24h period. As it is currently presented, the data are not correct and misleading

Please see response above.

  • Line 180: the use of the word “Dose” is not appropriate. Please use “intake” or something similar instead

The word ‘does’ has been replaced by ‘intake’ throughout the manuscript [Page 5, Lines 201, 205; Page 10, Line 378].

  • Line 208: please provide the equation used

The following equation from the mentioned reference in the manuscript has been added (lipid = 3.968 + [5.917 x creamatocrit value]) [Page 6, Lines 229-230].

  • Line 247: “data were” not “data was”

The has been corrected [Page 6, Line 254].

  • Line 252: “analyses” not “analysis”

This has been corrected [Page 6, Line 259].

  • Line 253: “were” not “was”

This has been corrected [Page 6, Line 260].

  • Statistics: in your models you present “week” as an explanatory variable which should indicate the effect of the intervention on your outcomes. However, the use of a diet quality measure such as HEI rather than “week” would provide much more insight into the potential relationship between maternal diet and HM composition

While we agree that this would be of interest, accurately measuring dietary intake is notoriously difficult and given the small sample size of this proof-of-concept study, we were concerned that this could affect the results. As a result, we chose to assess changes in parameters over time, and therefore used ‘week’ as the explanatory variable. We appreciate that the changes in dietary intake (and thus dietary quality) may account for the changes in hormone concentrations in HM, and in order to explore this further we have now undertaken further analyses to determine whether changes in hormone concentrations in HM were associated with changes in either maternal body weight/fat mass and/or changes in intake of specific nutrients [Page 10, Lines 367-374]. We consider that this is more informative than using a composite measure, such as HEI, since it can provide an indication as to whether there are specific nutrients that may be more important determinants of HM composition than others, although we appreciate that further more detailed studies are required.

  • Statistics: Please describe what descriptive statistics are presented (eg means SD or median IQR and did you use t-test or non-parametric tests)

The descriptive statistics presented included mean, standard deviation and range. This information has now been included in the revised manuscript [Page 6, Lines 260].

Results

  • Table 1: Maternal pre-pregnancy BMI and postnatal BMI was not different, suggesting that these women lost all of their pregnancy weight by the time they participated in the study. It not clear why this is the cohort chosen for a calorie-restriction/weight-loss intervention and would seem to be more clinically relevant and important in a population of women who had a BMI that indicated overweight and/or obesity. Please discuss the generalizabity and clinical importance of your findings considering the maternal BMI of this cohort.

Thank you for raising this point. The dietary intervention was focused on improving dietary composition/quality by reducing intakes of discretionary foods, saturated fats and added sugars rather than on weight-loss, and we aimed to recruit women who were keen to improve the quality of their dietary intake, rather than necessarily wanting to lose weight. Clearly, the shifts in dietary intake in this study were sufficient to produce significant reductions in maternal weight and fat mass over a 2 week period, which also enabled us to confirm that infant milk intake and infant growth were not affected, since there have been concerns raised that weight loss during lactation could compromise these parameters. In addition, the BMI of the women in this cohort covered a wide range (from 17.1 to 33.0kg/m2, with body weights from 54.4 to 80.7kg) and, since each woman was their own control across the study, our results suggest that improving dietary quality has effects on HM composition across a wide BMI range. Now that we have completed this initial proof-of-concept trial, we are obviously interested to understand whether more extreme and/or longer-term dietary interventions, particularly in women who are overweight or obese, produce similar effects on HM composition and, most importantly, whether there are any impacts on infant growth or body composition.

  • Table 2: Again it would be nice to see the change in dietary quality (e.g. HEI score)

We appreciate that overall dietary quality/degree of compliance with dietary guidelines would be an interesting addition to the study, we consider that our current approach (i.e. of considering each individual nutrient separately) provides more useful data, since it enables us to explore the impact of changes in specific nutrients. We absolutely agree that the HEI (or equivalent measures) will be extremely valuable as we move beyond the proof-of-concept studies into larger trials in specific (at-risk) populations.

  • Table 2: What statistical test was used? Where all data normally distributed?

All p-values were computed using univariate linear mixed effects models with individual mothers fitted as random effects in Table 3 (former Table 2), This information has now been added [Page 9, Lines 304-305].

  • Compliance to the dietary intervention should be assessed and reported in either the methods or the results

Compliance to the dietary intervention was assessed by contacting participants twice weekly via their preferred means and by incorporating all recorded foods/drinks consumed other than those provided as well as any of the provided foods that they did not consume into the dietary analysis [Page 3, Lines 124-127]. These measures indicated a high level of compliance to the dietary intervention. In addition, the significant reductions in body weight (-1.8%) and body fat mass (-6.3%) that was observed over the study period further support the compliance with the intervention. This information has been included in the manuscript [Page 12, Lines 460-463].

  • Table 3: What statistical test was used?

All p-values were computed using univariate linear mixed effects models with individual mothers fitted as random effects in Table 4 (former Table 3). This information has now been included [Page 9, Lines 319-320].

  • Line 307 – 308: milk production was not measured – infant volume intake of HM was measured

Please see response above.

  • Table 4: None of the models are discussed in the results section. Only the pre-post changes to maternal anthros and body comp. Please explain/present the models

Table 5 (former Table 4) has been updated and statistical models are described both in the methods section [Page 6, Lines 254-262] and in the table legend [Page 10, Lines 345-349].

You present data on the relationship between HM leptin and maternal body comp, but none of the other HM components? Also you do not mention if there was any associations at timepoints other than at baseline… Please present these data.

These were exploratory analyses with which we aimed to determine whether there was any evidence that 1) macronutrient and hormone concentrations were associated with maternal anthropometric measurements and HM production at baseline, and 2) if 24-h milk production and doses of fat, protein and lactose were associated with infant anthropometric measurements. The comparisons were only performed with values collected at baseline, since the primary objective was to determine whether (a) there was any evidence that HM composition was related to maternal anthropometric measures at the start of the study (as this has been reported in other studies) and (b) whether there was evidence that HM composition prior to intervention was associated with infant growth.

There was no association between any measures of maternal body composition and concentrations of HM adiponectin or insulin. This information has now been added to the revised manuscript [Page 10, Lines 363-365].

Also it is necessary that you model HM components with diet quality or the change in dietary components XYZ as a predictor in order to actually accomplish your aim of “determine the effect of reducing energy, fat and sugar intakes in lactating women over a 2-week period on the concentrations of macronutrients and metabolic hormones in HM”. As currently presented you have not actually completed your primary aim.

We agree that it is necessary to explore whether there were relationships between changes in dietary components and changes in hormone concentrations. These analyses have now been conducted and the change in HM adiponectin concentrations were positively associated with the changes in maternal carbohydrate (p=0.033) and total energy (p=0.038) intake. There were no associations identified between the change in HM insulin or leptin concentrations and changes in maternal fat and sugar intake. These results are reported in the revised manuscript [Page 10, Lines 371-374].

Further analyses have also suggested that the change in HM leptin concentrations was positively associated with the change in maternal body weight (p<0.001) and maternal fat mass (p<0.001), but there were no relationships between changes in either maternal body weight or fat mass and changes in levels of other hormones in HM [Page 10 , Lines 367-370]. While these data may suggest that the changes in HM adiponectin are more directly related to the change in diet composition, whereas changes in HM leptin are more related to the associated shift in maternal body weight/fat mass, we acknowledge that given the small sample size it is not possible to draw any clear conclusions and further studies are needed.

  • Line 333: milk production – again you did not measure this, you measured infant intake of HM

Please see above.

  • Line 334: use of the word “dose” please change to “amount”

This has been amended [Page 10, Line 380].

  • Was infant length correlated with fat intake across all weeks or just at one timepoint?

There was an error in text. Infant head circumference was positively correlated with the intake of fat consumed by the infant (p=0.026) across all weeks, not infant length [Page 10, Line 379]

Discussion

  • The study would be significantly strengthened if you could provide maternal circulating levels of hormones, as discussed Lines 384 -399

As indicated earlier, maternal blood samples were not collected in the study, since it was a proof-of-concept study and we were already performing a large number of measurements and human milk collections. We do agree that collection of maternal hormonal serum samples would have contributed to the study questions and would recommend inclusion of this parameter in future studies.

  • There needs to be some rationale for doing such a short intervention. Do you think this will have any positive long-term outcomes for the moms or will they go back to eating the way they were before?

As this was a proof-of-concept study, we estimated the timeframe required to identify changes in human milk composition based on limited previous literature and opted for an intermediate duration for the intervention. We reasoned that too short a period (i.e. a few days) would be insufficient for us to draw any clear conclusions, whereas if changes in composition were not observed within 2 weeks there was unlikely to be a rationale for a longer study.

The idea behind this study was never to test long-term outcomes or future changes in eating habits, particularly because we had provided all meals and snacks for the duration of the intervention which we understand would not be realistic for long-term maintenance. Rather, the outcome of the study would indicate the need for longer durations of dietary intervention. Rationale for the choice of dietary intervention timeframe has been added [Page 2, Lines 93-95].

  • The statement 417 -419 should be softened or edited to say that a “short-term reduction in energy intake” will not affect HM production or infant growth.

This statement has been updated [Page 12, Lines 465-468], and now reads:

“The results of the present study add to this previous literature, by suggesting that women can safely reduce energy intakes for a short-term during lactation without negative effects on milk production, HM macronutrient content or infant growth”.

Round 2

Reviewer 2 Report

The authors did an adequate job addressing the issues and should be considered for publication.

Author Response

Dear Hulda,

As requested, the manuscript has been revised, particularly in the methods section, to reference our previous publication to reduce repetition.

I would be happy to make further revisions if the match is still excessive based on the iThenticate report, but have tried to find a balance between directing the reader to previous publications and including information relevant to their understanding of the study design. 

Sincerely, 

A/Prof Bev Muhlhausler 
